# Changes of Flow and Sediment Transport in the Lower Min River in Southeastern China under the Impacts of Climate Variability and Human Activities

**Wen Wang** [1,*], **Tianyue Wang** [1], **Wei Cui** [1,2], **Ying Yao** [1], **Fuming Ma** [3], **Benyue Chen** [3] **and Jing Wu** [3]

[1] State Key Laboratory of Hydrology-Water Resources and Hydraulic Engineering Sciences, Hohai University, Nanjing 210000, China; hhuwty@hhu.edu.cn (T.W.); swcuiwei@hhu.edu.cn (W.C.); yaoying20212021@163.com (Y.Y.)

[2] Nanjing Hydraulic Research Institute, Nanjing 210000, China

[3] Bureau of Hydrology and Water Resources of Fujian Province, Fuzhou 350000, China; fjsstkcsjyxgs@163.com (F.M.); chenbenyuehhu@163.com (B.C.); bigmouth603@163.com (J.W.)

\* Correspondence: wangwen@hhu.edu.cn

**Abstract:** The Min River is the largest river in Fujian Province in southeastern China. The construction of a series of dams along the upper reaches of the Min River, especially the Shuikou Dam, which started filling in 1993, modified the flow processes at the lower Min River, leading to the significant increase in low-flows and slightly decrease in flood-flows. At the same time, reservoirs have more effects on the sediment transport process than flow process by trapping most sediment in the reservoirs, and greatly reduced the amount of sediment transporting downstream. Increase in vegetation cover also contributes to the decrease in sediment yield. The reduction in sediment together with excessive sand mining in the lower Min River resulted in the severe downward erosion of the riverbed. Using a reformulated elasticity approach to quantifying climatic and anthropogenic contributions to sediment changes, the relative contribution of precipitation variability and human activities to sediment reduction in the lower Min River are quantified, which shows that the sediment reduction is fully caused by human activities (including land use/land cover changes and dam construction).

**Keywords:** anthropogenic impact; hydrologic change; dam construction; suspended sediment; streamflow; precipitation; vegetation; elasticity approach





## 1. Introduction

It is known that the construction of reservoir dams will change the timing, magnitude, and frequency of low and high flows (Magilligan and Nislow, 2005; Wang et al., 2011) [1,2]. In addition, the sediment transported to downstream is reduced after dam construction. Globally, about 50% of the gross sediment flux was estimated being trapped in reservoirs (Nilsson et al., 2005) [3]. On an average, annual sediment fluxes of major large rivers in East, Southern, and Southeast Asian region reduced by more than 75% from 1960s to 2000s due to the effect of mega dams (Gupta et al., 2012) [4]. After the construction of the Three Gorges Dam (TGD) in China in 2003, the sediment discharge in the lower reaches of the Yangtze River decreased by 55% compared with 1993–2002, and 65% of the pre- to post-TGD decrease in sediment flux can be attributed to the TGD (Yang et al., 2015) [5]. Due to the effect of sediment trapping in the Lancang cascade dams, the sediment reduced by 74.1% in the Vietnamese Mekong Delta over a 55-year period (1961–2015) (Binh et al., 2020) [6]. In addition to the effects of dam construction, precipitation changes and vegetation changes resulted from climate changes (including changes of precipitation and temperature) may also have significant effects on runoff and sediment yield in the watershed (Song et al., 2016; Zhang et al., 2019) [7,8]. The reduction in sediment load would lead to riverbed erosion (Binh et al., 2020; Zheng et al., 2018) [6,9] river channel

adjustment (Xia et al., 2016) [10], salt tide intrusion in estuaries (Eslami et al., 2019) [11], and coastal area loss in river deltas (Luo et al., 2017; Liu et al., 2017) [12,13]. On the other hand, while all dams affect the hydrologic regime, the magnitude and type of impacts vary greatly by dam and location (Timpe and Kaplan, 2017) [14]. To quantify dam-induced hydrological impacts and quantify the relative contribution of climate and human effects to hydrological alteration across different geophysical regions would be helpful to build up a holistic view of human effects on hydrological dynamics and consequently helpful for sustainable sediment manage and future dam siting and operation.

The Min River is the largest river in Fujian Province of China with a series of major reservoirs built in the catchment, among which Shuikou Dam is the largest one. This reservoir controlled 86% of the drainage area of the Min River. In the present study, we will analyze how the hydrology in the lower reaches of the Min River, including the streamflow and the sediment transport, is altered under the joint effects of a series of dams, especially the construction of Shuikou Dam, and quantify the relative contribution of different factors to sediment load changes.

## 2. Data and Methods

### 2.1. Introduction about the Min River Basin

The Min River has a total length of 541 km and a total drainage area of 60,992 km$^2$. It originates from the Wuyi mountain range, flows from the west to the east, and pours into the East China Sea. The basin is mostly mountainous (Figure 1), and the average slope of the river is 5‰. The basin is dominated by the subtropical monsoon climate zone with a warm and humid climate. The average annual temperature is about 17–19 °C. The average annual precipitation is about 1600–1700 mm, declining gradually from upstream to downstream. The period during April to September is the flood season, and the runoff takes up about 75% of the total annual runoff. The maximum monthly runoff occurs in June when the main flood season in the river basin starts. From October to March of the following year, it is a low-flow season, and the average annual runoff reaches the minimum in January. The lower reaches of the Min River are often influenced by heavy typhoon rains from July to September, causing serious floods with high flood peaks and large water volume.

The Min River basin has very good vegetation coverage generally. According to GlobeLand30 land cover data at a 30 m resolution provided by National Geomatics Center of China (globallandcover.com), 74.5% of the basin is covered by forest. As a result, the sediment yield is very low, mostly below 0.14 kg/m$^3$ annually on average in the tributaries of Min River.

Several islands are present in the lower Min River channel near Fuzhou city. The river estuary is strongly affected by normal semi-diurnal tides. The maximum tidal range in the estuary is 7.04 m, and the average tidal range is 4.1 m.

Nine large reservoirs have been built in the Min basin, including Ansha Reservoir, Chitan Reservoir, Dongxi Reservoir, Gutianxi Reservoir, Shaxikou Reservoir, Shuikou Reservoir, Shuidong Reservoir, Jiemian Reservoir, and Jinzhong Reservoir (see Figure 1). Before 1970, only one major reservoir was built in 1959. During 1971–1990, 4 major reservoirs were completed. After 1990, four more reservoirs were completed. Among them, the Shuikou Reservoir, which is located in about 120 km above the Min River estuary, is the largest one. The construction of Shuikou Dam started in March 1987, and the filling of the reservoir began in March 1993. The reservoir has a maximum storage capacity of 2.6 billion m$^3$ and an effective storage capacity of 700 million m$^3$. The drainage area above the dam site is 52,438 km$^2$, accounting for 86% of the total basin area. The reservoir is mainly used for hydroelectricity, also used for flood control and navigation.

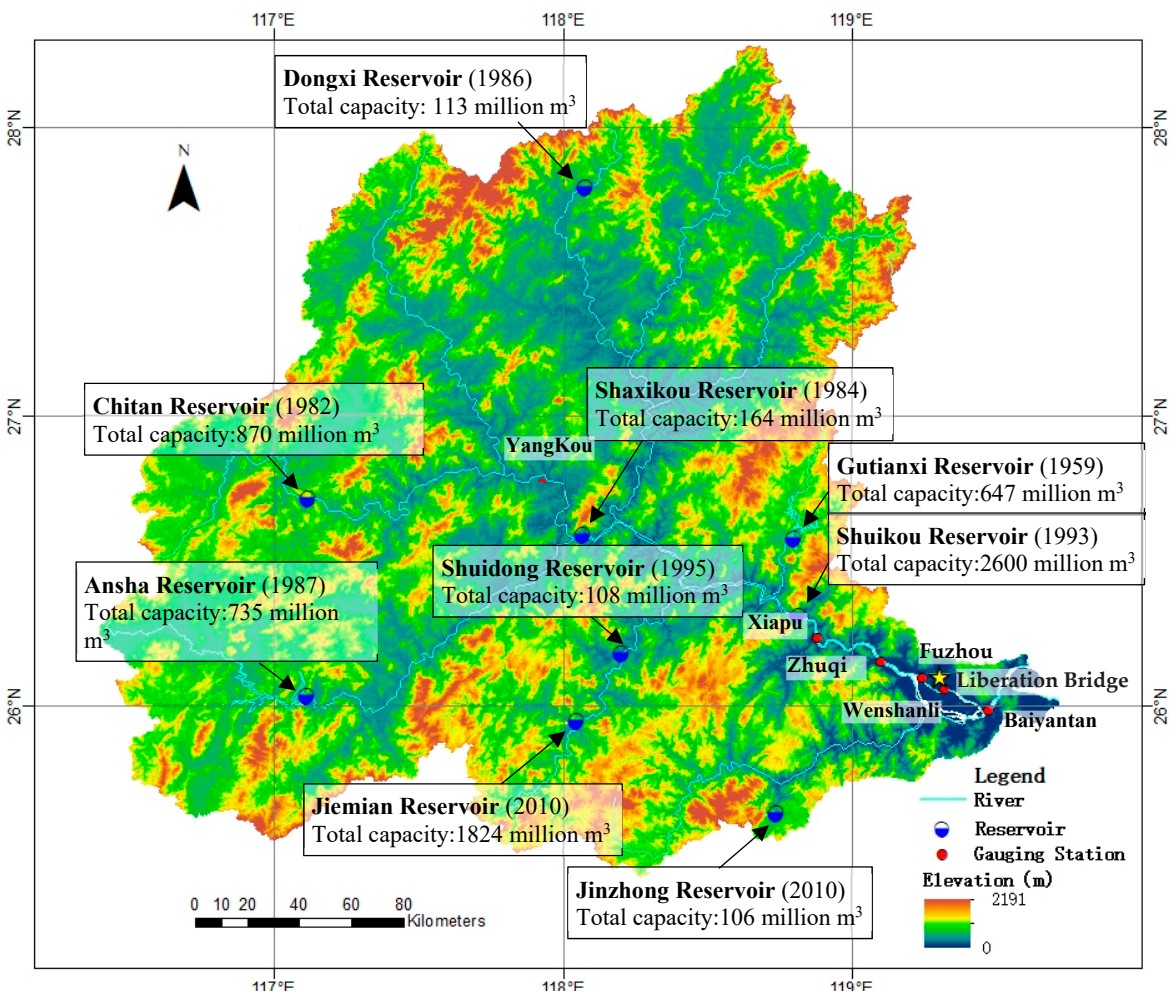

**Figure 1.** The elevation of the Min River basin and locations of major reservoirs and hydrologic gauging sites.

### 2.2. Data Used

Major hydrologic gauging stations located along the lower reaches of the Min River (below the Shuikou Dam) are displayed in Figure 1. In our study, the following hydrologic data are used:

(1)  Daily discharge data at Zhuqi station during 1950–2017. Zhuqi is the most important station along the lower reaches, located at about 45 km below the Shuikou Dam. It controls a drainage area of about 54,500 km$^2$. The observation of water level and river discharge at Zhuqi started in 1950;

(2)  Daily water level data at Xiapu, Zhuqi, Wenshanli, Liberation Bridge, and Baiyantan from 1950 to 2017 (except Xiapu, which started in 1993);

(3)  Daily suspended sediment concentration data at Zhuqi during 1952–2017;

(4)  Daily precipitation data at 19 precipitation gauging sites during 1950 (or 1951 or 1952) to 2017. The daily precipitation data are converted to the areal precipitation in Min River basin by the Thiesson polygon method.

The bathymetric data of the river channel along the cross-section at the Zhuqi in 2000, 2003, 2008, 2011, and 2014, and the thalweg data, which are derived from the lowest points of 79 cross-sections (about every 2 km along the main channel below Shuikou Dam) in 2011 and 2015 are used to analyze the change of river channel under the impact of Shuikou Dam operation. All the bathymetric data are provided by the Bureau of Hydrology and Water Resources of Fujian Province.

In addition, the GIMMS (Global Inventory Monitoring and Modeling System) third-generation NDVI (Normalized Difference Vegetation Index) data during 1981–2015 released by NASA (https://glam1.gsfc.nasa.gov) is used for analyzing the change of vegetation in the Min River basin.

*2.3. Methods*

2.3.1. Indicators of Flow Regime Changes

Richter et al. (1996) [15] proposed the RVA (Range of Variability Approach) method to evaluate the extent to which streamflow changed by human perturbations. They defined a suite of 32 ecologically relevant IHA (Indicators of Hydrological Alteration) parameters regarding the magnitude, timing, duration, frequency, and rate of water condition changes. In the present study, we investigate variations in hydrological conditions of not only the flow process (with focus on flood flows and low flows) but also the sediment transport process, therefore the following 10 indices are used to measure changes in the magnitude, timing, duration and frequency of flow and sediment conditions:

- The mean annual discharges (MAD)
- The annual coefficient of variation of daily discharges (CV)
- The annual minimum 7-day average discharge (Min7d)
- The annual maximum 1-day average discharge (Max1d)
- Julian date of each annual 1-day maximum (Dmax1d)
- Occurrence day of each annual 7-day minimum starting from August 1 (Dmin7d)
- The number of days with the discharge below the 15% percentile (N15p)
- The number of days with the discharge exceeding the 90% percentile (N90p)
- The mean annual concentration of suspended sediment (MSSC)
- The annual total load of suspended sediment (TSSL)

2.3.2. Methods of Change Detection

(1)  Exploratory change analysis

The double mass curve method (Searcy et al., 1960) [16] is commonly used to check the consistency of the relationship among different kinds of hydrologic data by comparing one series of data with another in the area. A double mass curve is a scatterplot of the cumulative sums of two variables. It exhibits a straight line when the relationship between the two keeps unchanged, and a break in the slope of the double mass curve indicates the change of their relationship.

(2)  Mann-Kendall trend test

The Mann-Kendall trend test (Kendall, 1975) [17], referred to as MK test hereafter, is used to test for the existence of possible trends in vegetation cover, precipitation, flow, and sediment transport processes. MK test is a rank-based nonparametric method, which is less sensitive to outliers than other parametric statistics. The MK method can test trends in a time series without specifying whether the trend is linear or nonlinear. We identify the type of temporal changes according to Kendall's $\tau$, which measures the strength of the monotonic trend, jointly with the $p$-value, which measures the level of significance, shown in Table 1.

**Table 1.** Classification of trend in terms of $p$-value and $\tau$ with Mann-Kendall (MK) trend test.

| $\tau$ | $p$-Value | Class | Trend Type |
|---|---|---|---|
| | $p < 0.01$ | +3 | Very significant increase |
| $\tau > 0$ | $0.01 \leq p < 0.05$ | +2 | Significant increase |
| | $0.05 < p \leq 0.1$ | +1 | Slight increase |
| $\tau = 0$ | $0.1 < p$ | 0 | No trend |
| | $0.05 < p \leq 0.1$ | −1 | Slight decrease |
| $\tau < 0$ | $0.01 \leq p < 0.05$ | −2 | Significant decrease |
| | $p < 0.01$ | −3 | Very significant decrease |

(3)    Pettitt test for change-point detection

The approach after Pettitt (1979) [18] is commonly applied to detect a single change-point in hydrological series or climate series. It tests the H0: no change, against the alternative: a change point exists. The non-parametric statistic is defined as:

$$K_T = \max|U_{t,T}|, \; 1 \leq t \leq T, \tag{1}$$

where the statistic $U_{t,T}$ is equivalent to a Mann–Whitney statistic for testing that the two samples $X_1, \ldots, X_t$ and $X_{t+1}, \ldots, X_T$ come from the same population, given by:

$$U_{t,T} = \sum_{i=1}^{t} \sum_{j=t+1}^{T} \mathrm{sgn}(X_i - X_j), \; t = 2, 3, \ldots, T, \tag{2}$$

where sgn($x$) = 1 if $x$ > 0, 0 if $x$ = 0, and −1 if $x$ < 0.

The significance probability associated with the value $K_T$ is approximately given by:

$$p \approx 2 \exp\left\{-6K_{t,T}^2 / \left(T^3 + T^2\right)\right\}. \tag{3}$$

The change-point of the series is located at $K_T$, provided that the statistic $K_T$ is significant at a given significance level (e.g., $\alpha$ = 0.05).

### 2.3.3. Quantification of the Attribution of Sediment Discharge Change

Generally, changes of suspended sediment load (SSL) can be attributed to climate variability and human activities. The elasticity approach proposed by Zhang et al. (2019) [19] is adopted and reformulated to quantify the effects of climate variability and human activities on changes of suspended sediment discharge in different periods, which are identified with abrupt-change detection methods.

Because the amount of sediment transport ($Q_S$) is the multiplication of the concentration of sediment ($C$) and streamflow discharge ($Q$), changes in $Q_S$ can be expressed in an elasticity form as (Zhang et al., 2019) [19]:

$$\frac{dQ_S}{Q_S} = \eta_Q \frac{dQ}{Q} + \eta_C \frac{dC}{C}, \tag{4}$$

where $\eta_Q = \frac{\partial Q_S / Q_S}{\partial Q / Q}$ and $\eta_C = \frac{\partial Q_S / Q_S}{\partial C / C}$ denote the elasticity of SSL to the streamflow and sediment concentration ~ streamflow ($C$–$Q$) relationships, respectively.

The reduction in $Q_s$ between two periods can be divided into two parts, i.e., the part due to changes in streamflow ($\Delta Q_{s,Q}$) and the part due to changes in the $C$–$Q$ ($\Delta Q_{s,C}$), which are given by

$$\Delta Q_{s,Q} = \sum_{j=1}^{n} q_j C_{j,1} \left(p_{j,2} - p_{j,1}\right) \tag{5}$$

$$\Delta Q_{s,C} = \sum_{j=1}^{n} q_j p_{j,2} \left(C_{j,2} - C_{j,1}\right) \tag{6}$$

where $q_j$ is the average daily discharge in class $j$ ($j$ = 1, $\ldots$ , $n$), which is one of the $n$ non-overlapping discharge classes with equal length of $1/n$ of the range of streamflow variation in log-space; $p_{j,1}$ and $p_{j,2}$ are the occurrence probability of discharge $q_j$ in two periods (i.e., period 1 and period 2) when the $C{\sim}Q$ relationship changed; $C_{j,1}$ and $C_{j,2}$ are the sediment concentrations for discharge $p_j$ in period 1 and period 2, respectively; $\overline{Q_S}$ is the average daily SSL; $\overline{Q_1}$, $\overline{Q_2}$, and $\overline{Q}$ denote mean values of daily discharge in period 1, period 2, and the whole period, respectively; $\overline{C_1}$, $\overline{C_2}$, and $\overline{C}$ denote mean values of sediment concentration in period 1, period 2, and the whole period, respectively.

Therefore, we have

$$\eta_Q = \frac{\partial Q_S / Q_S}{\partial Q / Q} \approx \frac{\left[\sum_{j=1}^n q_j C_{j,1}\left(p_{j,2} - p_{j,1}\right)\right] / \overline{Q_S}}{\left(\overline{Q_2} - \overline{Q_1}\right) / \overline{Q}} \tag{7}$$

$$\eta_C = \frac{\partial Q_S / Q_S}{\partial C / C} \approx \frac{\left[\sum_{j=1}^n q_j p_{j,2}\left(C_{j,2} - C_{j,1}\right)\right] / \overline{Q_S}}{\left(\overline{C_2} - \overline{C_1}\right) / \overline{C}}. \tag{8}$$

The relative change in $Q$ due to changes in climate (i.e., precipitation $P$, actual evaporation $E$, and potential evaporation $E_0$) and catchment property ($m$) is given by (Michael and Farquhar, 2011) [20]:

$$\frac{dQ}{Q} = \left[\frac{P}{Q}\left(1 - \frac{\partial E}{\partial P}\right)\right]\frac{dP}{P} - \left[\frac{E_0}{Q}\frac{\partial E}{\partial E_0}\right]\frac{dE_0}{E_0} - \left[\frac{m}{Q}\frac{\partial E}{\partial m}\right]\frac{dm}{m}, \tag{9}$$

with the respective partial differentials given by $\frac{\partial E}{\partial P} = \frac{E}{P}\left(\frac{E_0^m}{P^m + E_0^m}\right)$, $\frac{\partial E}{\partial E_0} = \frac{E}{E_0}\left(\frac{P^m}{P^m + E_0^m}\right)$ and $\frac{\partial E}{\partial m} = \frac{E}{m}\left(\frac{\ln(P^m + E_0^m)}{m} - \frac{P^m \ln P + E_0^m \ln E_0}{P^m + E_0^m}\right)$. The catchment property ($m$) can be estimated based on the catchment water balance equation and Budyko equation [21], which is described by:

$$Q = P - E - \Delta W \tag{10}$$

$$E = \frac{P E_0}{\left(P^m + E_0^m\right)^{1/m}} \tag{11}$$

where $\Delta W$ is the change in storage within the catchment which can be ignored in the long-term; $E_0$ is estimated using the FAO-56 method [22]. By combing Equations (10) and (11), we can estimate $m$ by solving the nonlinear equation.

By combining Equations (4) and (9), we have a comprehensive elasticity form for $Q_S$ change given as

$$\frac{dQ_S}{Q_S} = \eta_P \frac{dP}{P} + \eta_{E_0} \frac{dE_0}{E_0} + \eta_m \frac{dm}{m} + \eta_C \frac{dC}{C}, \tag{12}$$

with $\eta_P = \eta_Q\left[\frac{P}{Q}\left(1 - \frac{\partial E}{\partial P}\right)\right]$ and $\eta_{E_0} = -\eta_Q\left[\frac{E_0}{Q}\frac{\partial E}{\partial E_0}\right]$, denoting the elasticity of SSL to streamflow caused by variability of precipitation and $E_0$, respectively, and $\eta_m = -\eta_Q\left[\frac{m}{Q}\frac{\partial E}{\partial m}\right]$, denoting the elasticity of SSL to streamflow caused by changes of catchment property (such as topography, soil type/depth, geologic substrate, land cover, etc.). Here, $\eta_P$ and $\eta_{E_0}$ represent the effects of climate variability on SSL change, while $\eta_m$ and $\eta_C$ represent mostly the effects of human activities.

Changes in SSL attributed to changes in precipitation ($\Delta Q_{s,P}$), potential evapotranspiration ($\Delta Q_{s,E_0}$), and land surface (by changing streamflow, $\Delta Q_{s,m}$, and moderating $C\sim Q$ relationships, $\Delta Q_{s,C}$) can be, respectively, expressed as (Zhang et al. 2019) [19]:

$$\Delta Q_{s,P} = \eta_p \frac{\Delta P}{P} Q_s, \ \Delta Q_{s,E_0} = \eta_{E_0} \frac{\Delta E_0}{E_0} Q_s, \ \Delta Q_{s,m} = \eta_m \frac{\Delta m}{m} Q_s, \ \Delta Q_{s,C} = \eta_C \frac{\Delta C}{C} Q_s \tag{13}$$

The sum of $\Delta Q_{s,P}$ and $\Delta Q_{s,E_0}$ represents the contribution of climate variability to changes of sediment discharge, whereas the sum of $\Delta Q_{s,m}$ and $\Delta Q_{s,C}$ represents the contribution of human activities (including land use/cover change, dam construction, etc.). Therefore, the relative contribution of climate change and human activity are estimated by:

$$\varepsilon_{s,P} = \Delta Q_{s,P} / \Delta Q_s, \ \varepsilon_{s,E_0} = \Delta Q_{s,E_0} / \Delta Q_s, \ \varepsilon_{s,m} = \Delta Q_{s,m} / \Delta Q_s, \ \varepsilon_{s,C} = \Delta Q_{s,C} / \Delta Q_s \tag{14}$$

where $\Delta Q_s = \Delta Q_{s,P} + \Delta Q_{s,E_0} + \Delta Q_{s,m} + \Delta Q_{s,C}$.

### 3. Alteration of Streamflow and Sediment Process in the Lower Reaches of the Min River

*3.1. Changes of Streamflow Process*

The mean annual discharges (MAD) observed at Zhuqi during 1950 to 2017 are plotted in Figure 2a. The average annual discharge is 1718 m³/s during the whole 68 years, 1685 m³/s during 1950–1992, and 1775 m³/s during 1993–2017. Figure 2a shows no obvious overall trend during the whole 68 years. In comparison, the annual coefficient of variation (CV) of daily discharges exhibits a significant downward trend, as shown in Figure 2b.

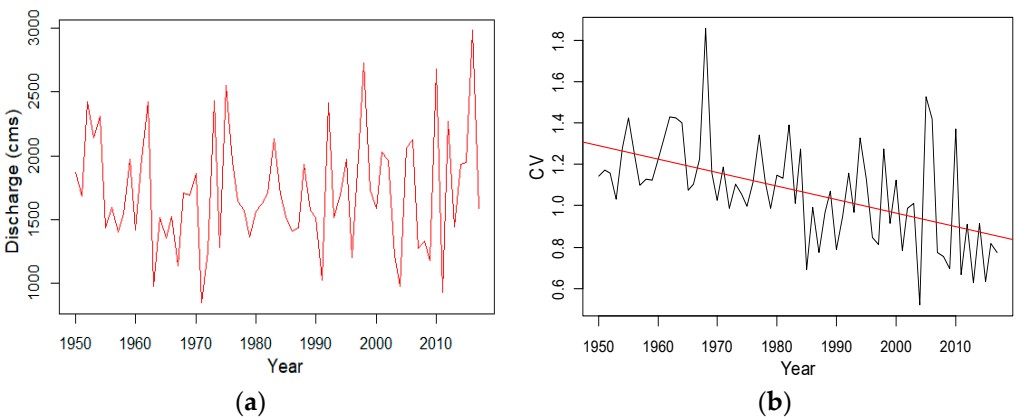

(**a**) (**b**)

**Figure 2.** Mean annual discharge (**a**) and annual coefficient of variation (CV) of daily discharge (**b**) observed at Zhuqi during 1950–2017.

The changes of low flows and flood flows during 1950~2017 are further investigated. The annual series of minimum 7-day average discharge (Min7d), maximum 1-day average discharge (Max1d), minimum 7-day average flow occurrence time (Dmin7d), maximum 1-day average flow occurrence time (Dmax1d), the number of days with the discharge below the 15% percentile (~510 m³/s) (N15p), and the number of days with the discharge exceeding the 90% percentile (~3470 m³/s) (N90p) are displayed in Figure 3. By visual inspection, we find an obvious positive trend in Min7d and an obvious negative trend in Max1d, at the same time, both N15p and N90p decreased. While other sequences showed no obvious changing trend.

Mann-Kendall trend test is applied to the annual flow, low flow, and flood flow series at Zhuqi. The test results are presented in Table 2, which show that MAD exhibits no significant changes, but significant negative trend is observed in the CV series; Min7D significantly increased, while the N15p significantly decreased; Max1D did not changed significantly although N90p significantly decreased. That is, the regulation of the reservoirs has no effect on total runoff although the increase in water surface due to the impounding of reservoirs may increase the evaporation slightly; reservoirs have significant impacts on low flows, but minor effects on flood flows. Overall, the regulation of reservoirs makes the runoff tend to be slightly more evenly distributed over the year, but because the effective capacity of reservoirs is far less than the total annual runoff, so the impacts on flood peak flow is not remarkable.

Table 2 also presents the results of Pettitt test, which indicate that some flow indices of Min River, including CV, Min7D, N15p, and N90p, experienced significant step changes in early 1980s'.

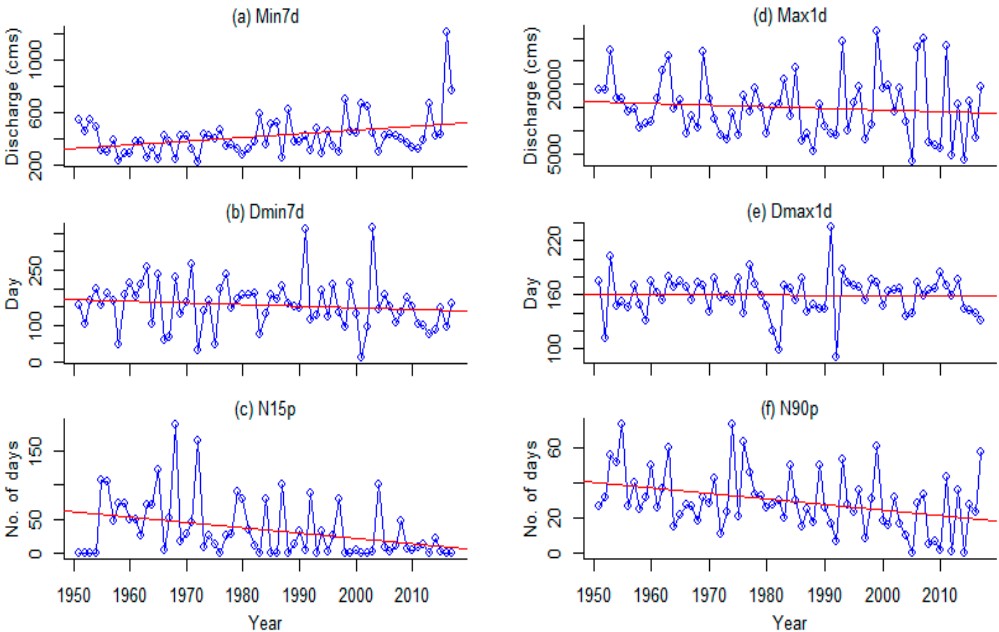

**Figure 3.** Changes of low flows and flood flows observed at Zhuqi during 1950 to 2017. (**a**) The annual series of minimum 7-day average discharge (Min7d); (**b**) minimum 7-day average flow occurrence time (Dmin7d); (**c**) the number of days with the discharge below the 15% percentile (~510 m³/s) (N15p); (**d**) maximum 1-day average discharge (Max1d); (**e**) maximum 1-day average flow occurrence time (Dmax1d); (**f**) the number of days with the discharge exceeding the 90% percentile (~3470 m³/s) (N90p).

**Table 2.** Trend and change-points in flow series detected by the MK test and Pettitt test.

| Data Type | Index | Trend | Year of Change |
|:---:|:---:|:---:|:---:|
| Flow | MAD | 0 | − |
| | CV | − ** | 1984 ** |
| | Min7d | + ** | 1981 ** |
| | Dmin7d) | 0 | − |
| | N15p | − ** | 1980 ** |
| | Max1d | 0 | − |
| | Dmax1d | 0 | − |
| | N90p | − ** | 1984 ** |
| Sediment | MSSC | − ** | 1993 ** |
| | TSSL | − ** | 1993 ** |

Note: + indicates positive trend; − indicates negative trend; ** indicates that the change is significant at 0.01 significance level.

The seasonal distribution of flow processes under conditions of different degree of reservoir impacts is investigated by comparing the average discharges in each day of the year (smoothed with 8 Fourier harmonics) during three different periods, i.e., 1950–1970, 1971–1992, and 1993–2017, in Figure 4a. For comparison, the variation of average precipitation in each day of the year during the three periods is plotted in Figure 4b. Comparing Figure 4a,b, the seasonal distribution of flow basically matches that of precipitation and did not changed much in the three periods by visual inspection. It is known that reservoirs generally control the water storage below a certain level before the major flood season (April to July in Min River basin) to prepare for the occurrence of major floods, then stores water after the major flood season, and finally increases the release of outflow in the low-flow period. However, because the effective storage capacity of all the 8 major reservoirs above Zhuqi station takes about only 7% of the average annual runoff of the Min River observed at Zhuqi, the impacts on flow processes are not obvious for high flows, but significant for low flows during November to February.

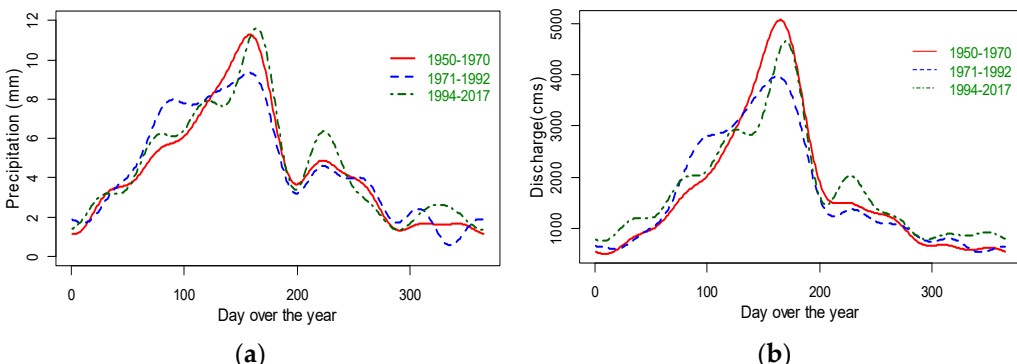

**Figure 4.** (**a**) Variation of areal precipitation above the drainage area above Zhuqi and (**b**) average daily discharge observed at Zhuqi during three different periods.

### 3.2. Variation in Sediment Transport

The annual average concentration of suspended sediment (SSC) observed at Zhuqi during 1952 to 2017 are plotted in Figure 5. According to the observation at Zhuqi, before the construction of Shuikou Dam, the average suspended sediment concentration was 0.129 kg/m$^3$ during the period from 1952 to 1992 with a maximum of 0.261 kg/m$^3$ in 1962 and the minimum of 0.065 kg/m$^3$ in 1991, and the average suspended sediment transport capacity was 7.15 million tons every year during 1951 to 1992. After the impoundment of Shuikou Dam in 1993, the average suspended sediment concentration at Zhuqi was 0.038 kg/m$^3$ during 1993–2017, with a maximum of 0.136 kg/m$^3$ in 2010 and a minimum of less than 0.007 kg/m$^3$ in 2008, and the average annual suspended sediment transport decreased to 2.48 million tons during 1993 to 2017. In comparison with the average of 7.15 million tons every year during 1951 to 1992, about 4.67 million tons (about 65%) suspended sediment was deposited in all the reservoirs above Zhuqi every year. By comparing the average of 6.06 million tons of sediment transport during the more recent period from 1983 to 1992 with the 4.67 million tons of sediment after 1993 when Shuikou Reservoir was built, it is estimated that about 3.58 million tons of suspended sediment, which would deposit in the lower Min River channel without the effect of reservoirs, was deposited in the Shuikou reservoir every year.

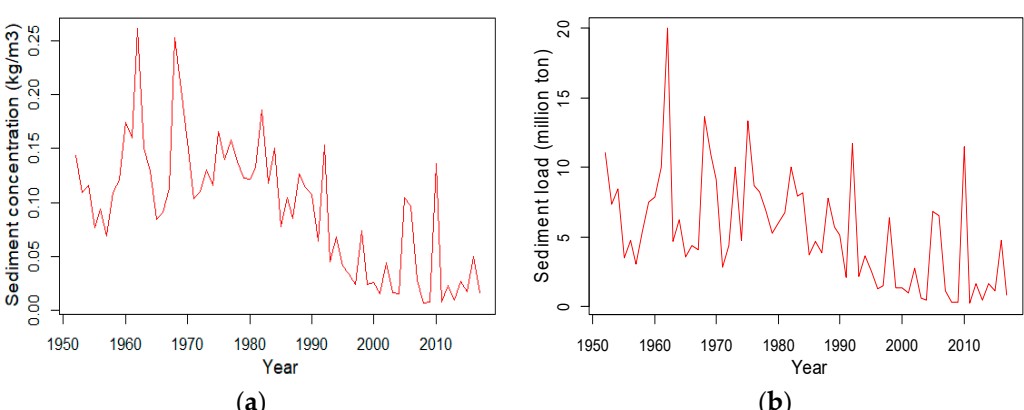

**Figure 5.** (**a**) Annual average suspended sediment concentration and (**b**) annual suspended sediment load observed at Zhuqi during 1952–2017.

According to the measurement at Zhuqi in 1990, bedload sediment took about 4.1~11.2% of total sediment when streamflow discharge was about 2000–8000 m$^3$/s, and the ratio of bedload to suspended load is about 4.3–12.6% with an average of 7.5%. As the average suspended sediment transport was 6.06 million tons at Zhuqi during 1983–1992 before the construction of Shuikou Reservoir, the average bedload sediment transport was about 0.455 (≈6.06 * 7.5%) million t. Xu et al. (2012) [23] estimated that the average annual

bedload sediment below the Three Gorges Dam during 2003~2010 after the impoundment of the reservoir in 2003 decreased by 93% in comparison with that during 1991–2002. As the Shuikou dam controls 96.2% of the total drainage area above Zhuqi, it is estimated that after the completion of Shuikou reservoir, about 0.41 million tons ($\approx$ 7.5% * 93% * 96.2% * 6.06 million t) of bedload sediment was deposited in the Shuikou reservoir every year due to the reservoir's retention. Therefore, after the completion of Shuikou reservoir, an average of 3.99 million t of sediment (including 3.58 million t of suspended load and 0.41 million t of bedload) was deposited in the reservoir area every year compared with the 10 years before the completion. Assuming a sediment density of 1300 kg/m$^3$, the Shuikou dam area loses about 3.07 million m$^3$ of its capacity per year due to sediment deposition, which means the reservoir loses about 1.2% of its initial capacity (2.6 billion m$^3$) every 10 years.

The double mass curve method is further applied to the investigation of the change point detection for the consistency in the relationship between runoff and sediment load. We plot the double-mass curves of cumulative annual runoff versus cumulative annual SSL at Zhuqi in Figure 6a and cumulative runoff in June versus cumulative sediment load in June in Figure 6b. It is shown that there is a slope break in 1993 in Figure 6a, indicating the annual total amount of sediment significantly changed in 1993, but in June, when the runoff and sediment are the most over the year, significant changes occurred in 1984 (see Figure 6b). The year 1993 is when the largest reservoir, Shuikou dam, started the impoundment, whereas 1984 is the year when the Shaxikou dam, which is the second largest reservoir in the drainage area (25,562 km$^2$) in the Min River Basin, started its cofferdam construction.

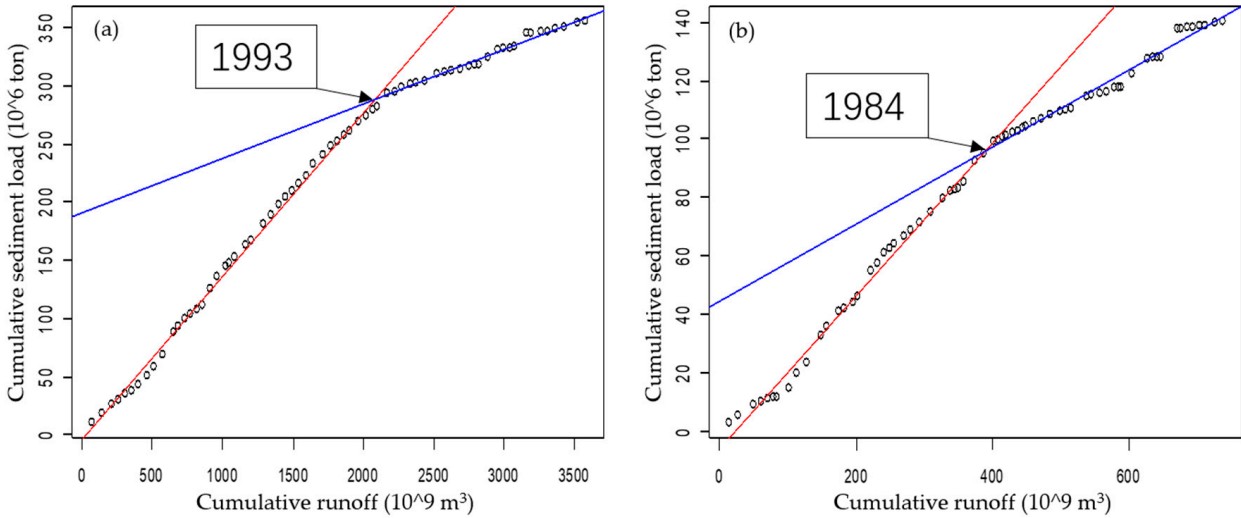

**Figure 6.** The double-mass curve of (**a**) cumulative annual runoff versus cumulative annual suspended sediment load and (**b**) cumulative runoff in June versus cumulative sediment load in June at Zhuqi.

The daily streamflow and sediment concentration~streamflow (C~Q) relationship with 300 non-overlapping discharge classes between the period 1952–1992 (Period-I) and the period 1994–2018 (Period-II) observed at Zhuqi in the Min River are plotted in Figure 7. A visual inspection of Figure 7 indicates the significant reduction in sediment concentration for any water discharge classes.

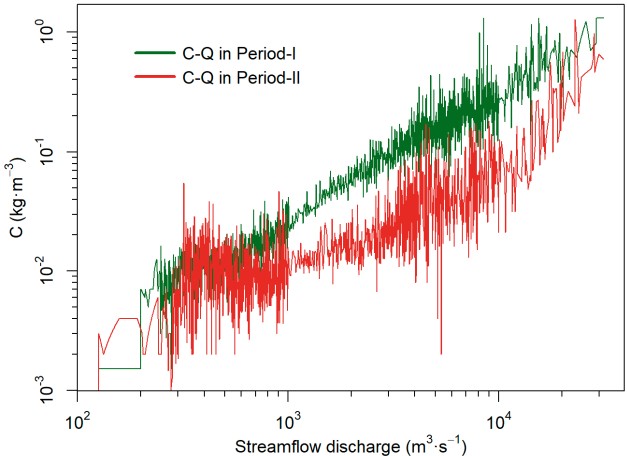

**Figure 7.** The sediment rating curves between suspended sediment concentration (*C*) and streamflow discharge with 300 discharge classes during the period 1952–1992 (Period-I) and the period 1994–2018 (Period-II) observed at Zhuqi in the Min River.

### 3.3. *Changes of Water Level and Riverbed along the Lower Min River below Shuikou Dam*

Due to the changes in flow regime, especially in sediment discharge, jointly with the influence of human sand mining activities, the shape of the riverbed in the lower reaches of the Min River is constantly changing. Such kind of changes is especially exhibited in the downcutting in the thalweg of lower Min River.

After the Shuikou Reservoir was completed, the thalweg at many river cross-sections along the lower reaches of the Min River got deepened generally and oscillated to varying degrees. As shown in Figure 8, the maximum incision depth in the riverbed during 2000 and 2015 exceeds 20 m at Zhuqi. Figure 9 shows the changes of the thalweg of the lower reaches of the Min River from the Shuikou Dam to the estuary in 2011 and 2015.

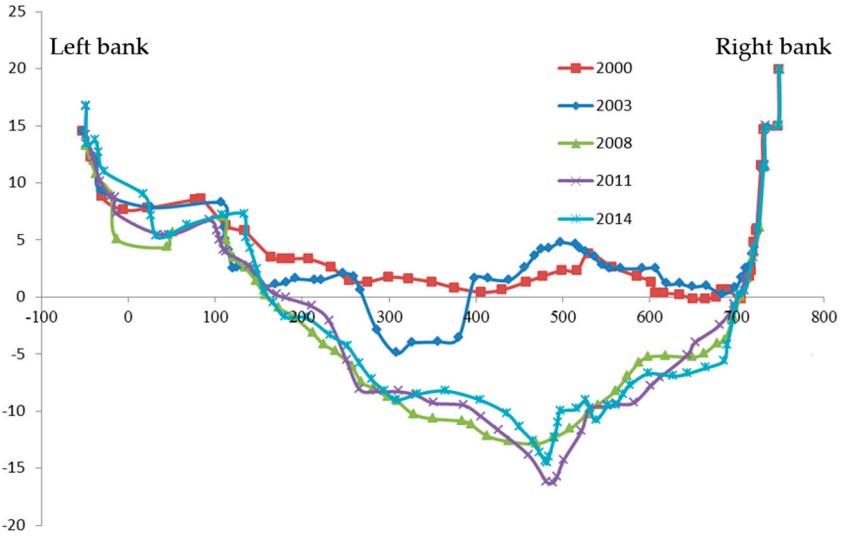

**Figure 8.** River bed elevation at the cross-section of the Min River at Zhuqi in different years.

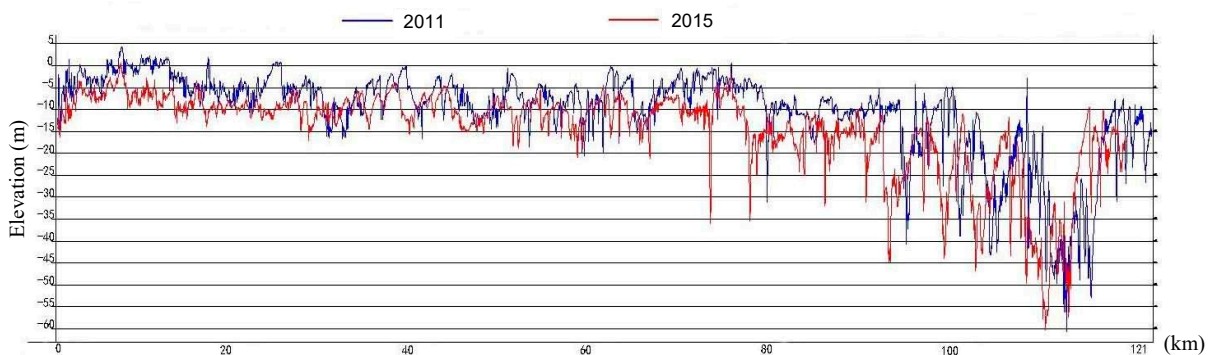

**Figure 9.** Comparison of thalwegs of the lower Min River (start from the Shuikou Dam to the river mouth) in 2011 and 2015.

River channel erosion below dams is a common phenomenon all over the world (ICOLD, 2009) [24]. Similar to the impact of the construction of the Shuikou Reservoir in the Min River, many examples of serious channel incision below dams in relation to regional geographic conditions, e.g., 6 m downcutting below Davis Dam, Colorado River, USA (Mathias, 1997) [25]; 7.5 m downcutting below Hoover Dam, Colorado River, USA (ICOLD, 2009) [24]; 6 m downcutting below Saulspoort Dam, Ash River, South Africa (ICOLD, 2009) [24]; and 10 m downcutting below Three Gorges Dam, Yantze River, China (Zheng et al., 2018) [9]. However, bed erosion is not the only possibility and local conditions may determine a variety of outcomes (Lai et al., 2017) [26] in terms of the incision depth or the time to reach a new balance between erosion and deposition. A recent study showed that the new long-term hydro-morphological equilibrium may have been established in the middle and lower Yangtze about 14 years after the closure of TGD (Lai et al., 2017) [26]. In the case of the Shuikou Dam, such a new equilibrium is still yet to be achieved. Although there is not too much cutting down along the cross-section at Zhuqi during 2008 to 2014 (as shown in Figure 7), indicating that the overall elevation of the riverbed got more or less stable at Zhuqi, the thalweg is still getting deeper generally along the lower reach of Min up to 2015 (as shown in Figure 8). We can deduce that the riverbed is still in a process of downward erosion since the Shuikou Dam has been built for more than 20 years.

The lowest water level at several locations along the lower reach of the Min River (see Figure 10) shows that the lowest water level of the lower reach has decreased significantly since late 1980s', except at the mouth of the river (Baiyantan). The decline of the water level accelerated in the late 1990s.

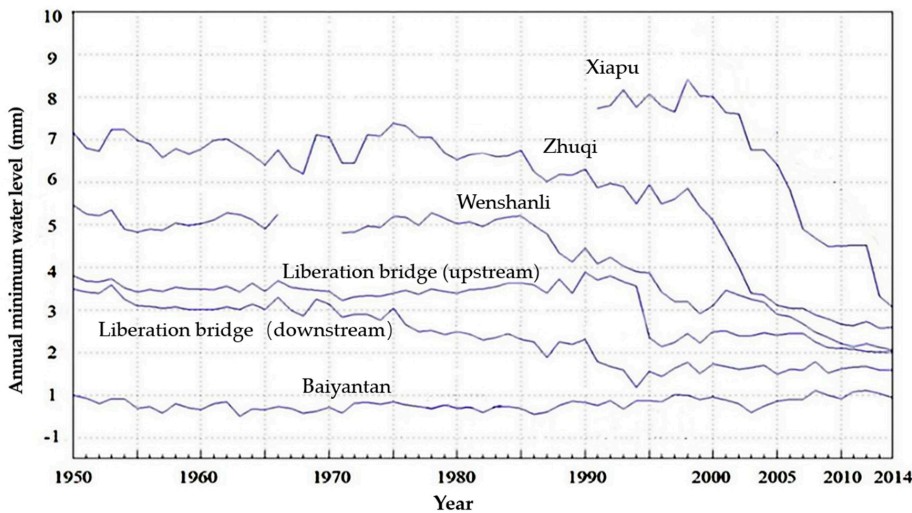

**Figure 10.** The variation of the lowest water level along the lower reach of the Min River.

The stage-discharge rating curves at Zhuqi station in 1950, 1970, 1990, 2000, 2010, and 2017 are plotted in Figure 11. A comparison of these curves shows that the water level with a specific discharge changed very little during 1950 to 1970, but dropped significantly after late 1990s. With a discharge of 2000 m$^3$/s, the average water level was 8.28, 8.3, 7.73, 6.93, 3.73, and 3.60 m in 1950, 1970, 1990, 2000, 2010, and 2017, respectively, that is, the average water level dropped 4.68 m in 68 years and such a drop mostly occurred in the last 30 years.

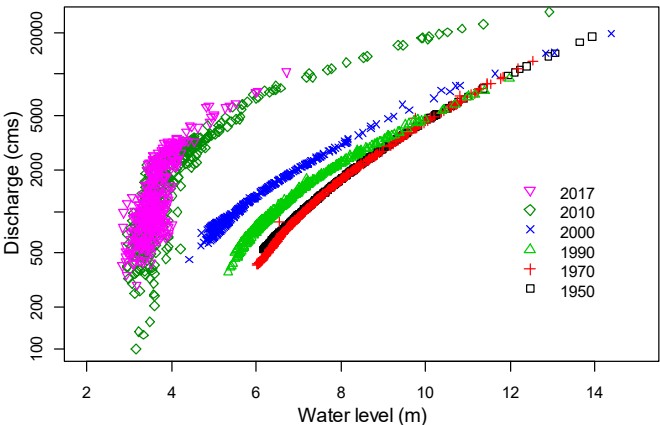

**Figure 11.** The stage-discharge rating curves at Zhuqi station in different years.

## 4. Contributing Factors of Sediment Reduction

River sediment comes from many sources including the erosion from land surfaces, stream banks and stream beds. Although the runoff is the direct driving force of transport sediment in the river, the fundamental driving force of the erosion comes from the precipitation. Therefore, changes of the amount and intensity of precipitation, land coverage of vegetation, and the construction of a series of dams may result in changes of sediment flux in the river.

### 4.1. Precipitation

First of all, we calculated the areal annual precipitation and the annual runoff coefficient in the drainage area above Zhuqi Station, as shown in Figure 12. It can be seen that the annual precipitation of Min River basin increased slightly in the period from 1950 to 2017, while the annual runoff coefficient decreased slightly. However, MK trend test and Pettitt change point test found that these changes were not significant. Further comparison of the correlation between annual precipitation and annual runoff before and after 1993 found that (see Figure 13), after 1993, the slope of the linear regression line between annual runoff and annual precipitation gets smaller, indicating that the annual runoff is less sensitive to the variation of precipitation or, in another word, the variation of annual precipitation results less variation of annual runoff. This implies that a large number of large reservoirs in the Min river basin have the capacity of modulate the variation of annual runoff by reducing the outflow in wet years and increasing the outflow in dry years.

The change of precipitation process affects not only runoff process but also sediment yield. The change of precipitation has strong effects on sediment yield. With the increase in rainfall intensity, the erosion capability of rainfall increases, which may lead to the increase in soil loss. According to the USLE soil loss equation (Wischmeier, 1959) [27], one rainfall event is considered to be erosive if the total precipitation amount $p \geq 12.7$ mm or the amount of precipitation in 15 min $p\,15 \geq 6.4$ mm. However, the standards of erosive rainfall vary from region to region. In China, the rainfall amount varies between 9 and 14 mm (Xie et al., 2000) [28]. Huang et al. (2015) [29] investigated the rainfall erosivity in Changting County in Fujian Province and found that the criterion of erosive rainfall was $p \geq 13$ mm per day. We investigated the maximum 1-day precipitation and the total amount of erosive precipitation with daily rainfall $p \geq 13$ mm from 1950 to 2017 observed at Yangkou in the center of Min River basin and Zhuqi in the lower part of Min River basin.

As shown in Figure 14, the maximum 1-day precipitation and the total erosive precipitation of Yangkou and Zhuqi showed no significant changes, and MK trend test and Pettitt change point test also confirmed no significant changes. That is to say, the change of Min river sediment discharge has very little to do with the change of precipitation.

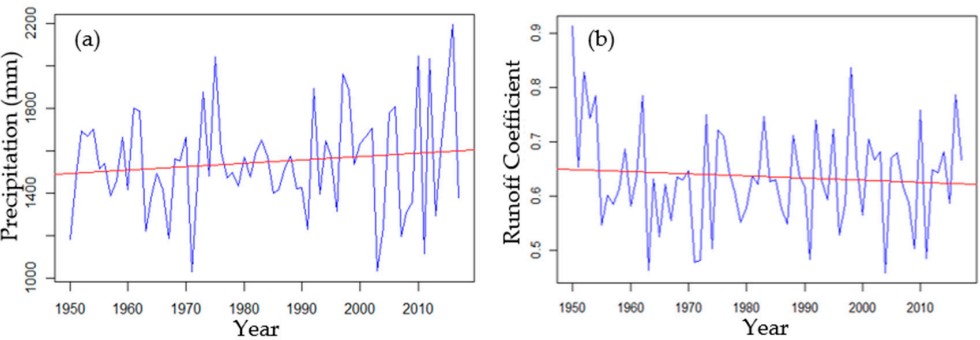

**Figure 12.** Variation of (**a**) annual precipitation and (**b**) runoff coefficient above Zhuqi.

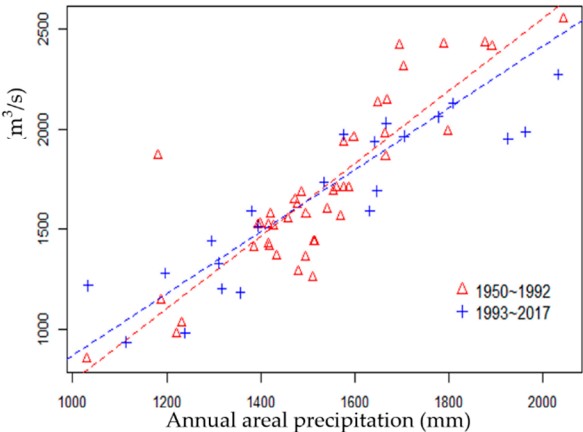

**Figure 13.** The correlation between annual areal precipitation and annual runoff at Zhuqi in the period before and after 1993.

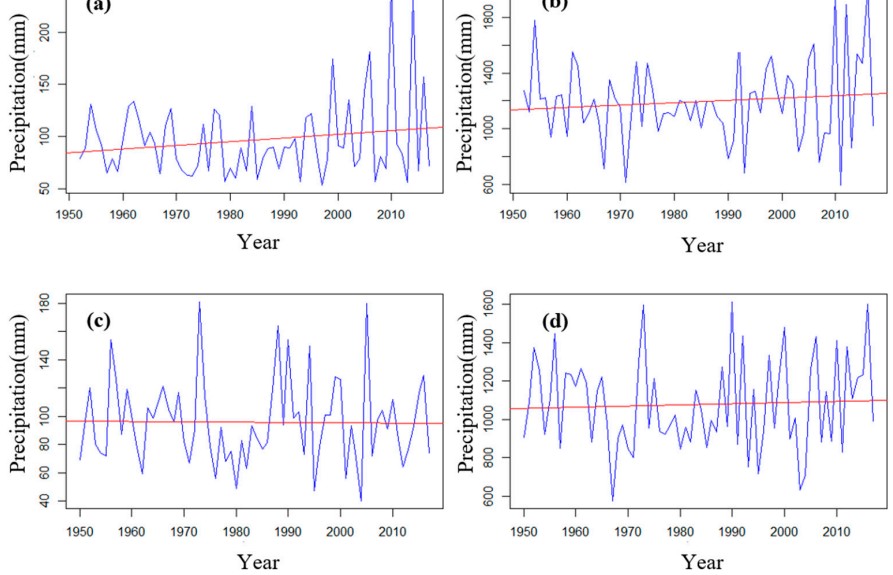

**Figure 14.** Variation of the maximum 1-day precipitation and the total erosive precipitation of Yangkou and Zhuqi. (**a**) Maximum 1-day precipitation of Yangkou, (**b**) total erosive precipitation of Yangkou, (**c**) maximum 1-day precipitation of Zhuqi, and (**d**) total erosive precipitation of Zhuqi.

## 4.2. Vegetation

The changes of sediment volume and sediment grain size is not only affected by the operation of reservoirs but the increase in vegetation cover over the basin may also play an important role in the variation of sediment generation process in the Min River basin. Using NASA's GIMMS (Global Inventory Modeling and Mapping Studies) NDVI (normalized difference vegetation index) data product during 1981–2015 (https://glam1.gsfc.nasa. gov), we calculated the annual areal average NDVI over the Min River basin, plotted in Figure 15a, which shows an obvious upward trend. Mann-Kendall trend test is also conducted for monthly NDVI on a grid basis, and the result is presented in Figure 15b. It is clearly shown that the increase in NDVI is very significant over most parts of the Min River basin. Such kind of significant increase in vegetation cover inevitably could lead to the decrease in sediment yield because vegetation restoration can effectively prevent soil loss (Wang et al., 2016) [30].

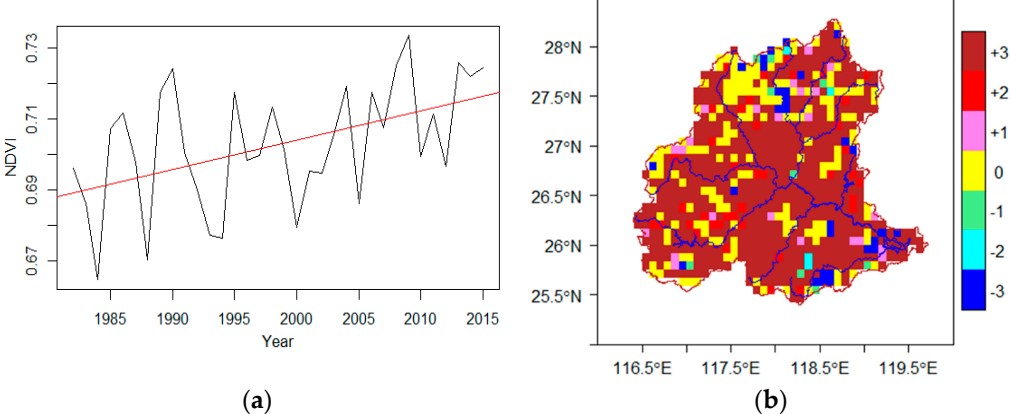

(**a**)                                                       (**b**)

**Figure 15.** Changes of the annual areal average NDVI (**a**) and the trend of monthly NDVI (**b**) over the Min River basin during 1981 to 2015. (Note: the number +3, +2, +1, and 0 stand for very significant trend, significant trend, weak trend, and no trend, respectively, and the sign (+ or −) stands for negative or positive trend.).

## 4.3. Dam Construction

The Min river is rich in runoff, but the runoff varied seasonally due the effects of monsoon climate. To adjust the seasonal distribution of runoff, and meanwhile take advantage of the rich hydropower in the river basin, numerous reservoirs and hydropower plants have been constructed, including nine large reservoirs with capacity over 100 million m$^3$ as shown in Figure 1. Figure 16 shows the significant increase in the total reservoir capacity of the reservoirs in the Min River basin during 1954 to 2011.

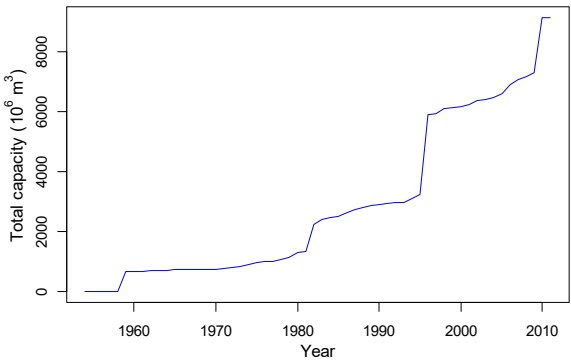

**Figure 16.** The change of total reservoir capacity in the Min River basin during 1954–2011.

The effects of dams on sediment generation and transport are mixed. After the impoundment of a reservoir, more sediment will deposit in the reservoir area, resulting in the reduction in sediment transport to the downstream. At the same time, due to the decrease in sediment concentration in the streamflow, the erosivity of streamflow will increase, which leads to more intensive erosion of the downstream river channel. However, in general, the sediment reduction caused by the sedimentation in reservoirs is much greater than the sediment increase caused by downstream channel erosion. For example, the curve of cumulative annual runoff versus cumulative annual sediment transport at Zhuqi in Figure 6 shows that the abrupt decrease in annual sediment transport in 1993 and the abrupt decrease in sediment transport during the main flood season in 1984 are respectively related to the construction of Shuikou reservoir and Shaxikou reservoir, which are the first and the second in drainage area in the Min River basin. On the other hand, the construction of other major reservoirs above Zhuqi have no significant impacts on the SSL at Zhuqi. That means, the significant sediment reduction is mostly caused by those large reservoirs, which take large portions of the drainage area.

### 4.4. Sand Mining

Sand mining is another major cause of sediment reduction. There is a huge demand for river sand, and large-scale sand mining occurred in the lower reaches of the Min River. There is no accurate data about the amount of river sand mining because many sand mining operations are illegal. The annual river sand consumption in the lower Min is estimated to be more than 10 million tons, which is much more than the sediment transported to the lower Min from the upstream reaches in recent years after the construction of Shuikou dam in 1993.

When sand mining removes huge amount of sediment, significant river bed incision is inevitable (e.g., Hackney et al., 2020) [31]. Figure 8 shows the dramatic deepening of the channel by over 10 m between 2003 and 2008 at Zhuqi, whereas the bathymetric changes between 2000–2003 and 2008–2014 are not so significant compared to the change between 2003 and 2008. Moreover, the annual average SSC and annual SSL on Figure 5 show no indication of huge reduction between 2003 and 2008. Therefore, we can deduce that sand removal by mining activities plays a very important role in the deepening of the lower Min River channel, especially between 2003 and 2008.

### 4.5. Relative Attribution of Sediment Discharge Changes

The change of areal annual precipitation observed in the drainage basin above Zhuqi and annual average streamflow discharge, average suspended sediment concentration, and suspended sediment load observed at Zhuqi during period-1 (before 1993) and period-2 (after 1993) are summarized in Table 3. Here, $E0$ is estimated using the meteorological variables observed at 19 sites within the Min River basin with the FAO-56 method (1976); the catchment property ($m$) is estimated using the Equations (10) and (11). According to the statistics presented in Table 3, we know that the changes of precipitation, $E0$, and m between period-1 and period-2 are very small (3%, −0.5%, and −3.4%, respectively), whereas the change of sediment concentration ($C$) is considerable (−70.5%). For a given catchment, many of the catchment properties (such as topography, soil type/depth, geologic substrate, etc.) remain nearly constant except for the land cover. Therefore, the change in the value of $m$ mostly reflect the change of land cover, and a change in vegetation cover from grasses to trees would increase $m$ (Roderick and Farquhar, 2011) [17]. However, the estimated $m$ for the Min River basin decreased in the period 2, which seems to be in contradiction to the fact that the average NDVI significantly increased during 1981~2015 (as shown in Figure 15). Such a contradiction likely resulted from the uncertainty in the estimation of areal precipitation and E0.

**Table 3.** Changes of areal annual precipitation (*P*), potential evaporation (*E*₀), and catchment property (*m*) in the drainage area above Zhuqi, and annual average streamflow discharge (*Q*), average suspended sediment yield (*Q_S*), and suspended sediment concentration (*C*) observed at Zhuqi during period-1 (1951–1992) and period-2 (1993–2017).

| *P* (mm/year) | | *E*₀ (mm/year) | | *Q* (mm/year) | | *Q_S* (t/km²/year) | | *C* (kg/m³) | | *m* (−) | |
|---|---|---|---|---|---|---|---|---|---|---|---|
| $P_1$ | $\Delta P$ | $E_{0,1}$ | $\Delta E_0$ | $Q_1$ | $\Delta Q$ | $Q_{S,1}$ | $\Delta Q_S$ | $C_1$ | $\Delta C$ | $m_1$ | $\Delta m$ |
| 1653.7 | 50.6 | 1087.9 | −5.4 | 973.5 | +60.3 | 131.2 | −85.7 | 0.129 | −0.091 | 1.056 | −0.037 |

Note: the subscript "1" indicates the value in period 1; Δ is the value in period 2 minus the value in period 1.

The elasticity indices $\eta_Q$ and $\eta_C$ for the Min River are computed using Equations (7) and (8) for different number of discharge classes (*n*), and the results are plotted in Figure 17. For *n* approximately greater than 800, the variations of $\eta_Q$ and $\eta_C$ are generally stable, and we take $\eta_Q = 0.95$ and $\eta_C = 1.09$ at *n* = 1000 as the representative values of $\eta_Q$ and $\eta_C$.

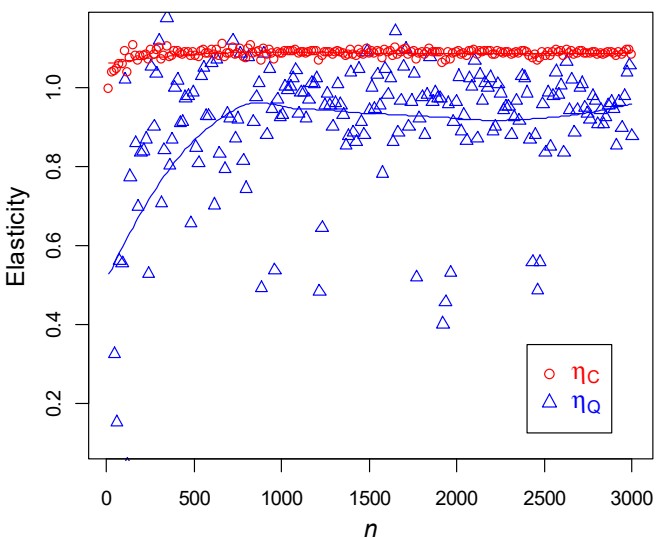

**Figure 17.** Variation of elasticity indices $\eta_Q$ and $\eta_C$ for the Min River vary with increasing *n*.

Then we can calculate the elasticity of suspended sediment load (*Qs*) to *P*, *E*0 and *m* with Equation (12), and the relative contribution of different contributing factors with Equation (14). The results are presented in Table 4. The absolute values of the elastic indices in Table 4 showed that $\eta_P > \eta_C > \eta_m > \eta_{E0}$, indicating that the sensitivity of *Qs* to these variables can be ranked as precipitation > sediment concentration (*C*) > catchment property (*m*) > potential evapotranspiration (*E*0). Although *C* is not the most influential factor to *Qs* change, it changed the most between period-1 and period-2, whereas there is little change in precipitation, *E*0, and *m*, hence the contribution to *Qs* reduction is dominated by the change of *C*. The sum of $\varepsilon_{s,m}$ and $\varepsilon_{s,C}$, which represents the contribution of human activities (including land use/cover change and dam construction), is estimated to contribute 103.8% of the reduction in suspended sediment, whereas climate variability only contributed −3.8% to the reduction in suspended sediment.

**Table 4.** Elasticity of suspended sediment load (*Qs*) and contribution to *Qs* reduction with respect to precipitation (*P*), potential evapotranspiration (*E*0), sediment concentration (*C*), and catchment property (*m*).

| Elasticity of $Q_s$ | | | | | Contribution to $Q_s$ Reduction (%) | | | |
|---|---|---|---|---|---|---|---|---|
| $\eta_Q$ | $\eta_P$ | $\eta_{E0}$ | $\eta_m$ | $\eta_C$ | $\varepsilon_{s,P}$ | $\varepsilon_{s,E0}$ | $\varepsilon_{s,m}$ | $\varepsilon_{s,C}$ |
| 0.95 | 1.34 | −0.39 | −0.41 | 1.09 | −3.6 | −0.2 | −1.3 | 105.1 |

## 5. Conclusions

Although there was no significant change of precipitation during 1950–2017, the construction of a series of dams along the upper reaches of the Min River, especially the Shuikou Dam, which started filling in 1993, modified the flow processes, leading to the significant increase in low-flows and slightly decrease in flood-flows at Zhuqi located at the lower Min River. At the same time, the variability of annual runoff is less sensitive to the variability of precipitation because large reservoirs can modulate the variation of annual runoff by reducing the outflow in wet years and increasing the outflow in dry years. Reservoirs have more effects on the sediment transport process than flow process by trapping most sediment in the reservoirs, and greatly reduced the amount of sediment transporting downstream. Vegetation over the Min River basin increased significantly in the last four decades, and the increase in vegetation cover also contributes to the decrease in sediment yield. The reduction in sediment together with excessive sand mining in the lower Min River resulted in the severe downcutting of the riverbed. The average water level observed at Zhuqi dropped 4.68 m during 1950 to 2017, and such a drop mostly occurred after the construction of Shuikou Dam. Using a reformulated elasticity approach to quantifying climatic and anthropogenic contributions to sediment changes, the relative contribution of precipitation variability and human activities to sediment reduction in the lower Min River are quantified, which shows that the sediment reduction is fully caused by human activities (including land use/land cover changes and dam construction).

**Author Contributions:** Conceptualization and methodology, W.W.; calculation and analysis, T.W. and W.C.; data collection and processing, B.C., Y.Y., and J.W.; project administration, F.M.; writing, all authors. All authors have read and agreed to the published version of the manuscript.

**Funding:** This research was funded by National Science Foundation of China project (grant number 419710420) and Water Conservancy Science and Technology Project of Fujian Province.

**Institutional Review Board Statement:** Not applicable.

**Informed Consent Statement:** Not applicable.

**Conflicts of Interest:** The authors declare no conflict of interest.

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
