# Peer review of "Changes of Flow and Sediment Transport in the Lower Min River in Southeastern China under the Impacts of Climate Variability and Human Activities"

_water, doi:10.3390/w13050673_

Round 1

Reviewer 1 Report

Strong points of the work

The problem addressed is very interesting and current in many countries. The analysis of the available data was done with great skill and with appropriate methods. The state of art is thorough and complete. The work is organized sequentially and easy to read even for the less experienced. The topic addressed is of both scientific and technical interest.

Weak points of the work

The data analysis was done with appropriate but not always innovative procedures. The work carried out is closer to a technical report than to a scientific publication with original content.

Some questions for the authors, if this is possible and compatible with the review work. I hope these questions are useful for further insights and work ideas.

  1. Were the dam construction projects preceded by an environmental impact analysis?
  2. If so, have the forecasts made been respected?
  3. If they have not been respected, what to attribute the errors to?
  4. What are the solutions to reach a condition of equilibrium for the best management of the water system?
  5. Are the authors sure that the removal of aggregates for the construction of road and rail embankments was not an important cause of river erosion?

Reviewer 3 Report

A quite interesting paper, with international relevance.

Please see the attachment with some minor improvements/comments.

Best wishes

Round 2

Reviewer 2 Report

The revision has been greatly improved from the previous version. I have a few comments.

  1. Lines 49-52 The sentence requires rewriting in two ways. First, the hydrology is altered, not altering, in the lower reach of the Min River. Secondly, the change occurred in the lower reach of the Min River due to the construction of dams does not affect the hydrology from the dam since there is no feedback mechanism. In addition, the introduction can be further improved by defining the research problem better.
  2. The authors stated that flow and sediment have not reached equilibrium in Abstract and Conclusions. How did you draw such a conclusion? Based on Figures 8 and 9? They just show two different shapes in two periods quite after the construction of the dam. If it is based on Figure 10, please extend the year to 2020, not ending at 2014.
